



**Decennial time trends and diurnal patterns of particle number**
**concentrations in a Central European city between 2008 and 2018**
Santtu Mikkonen[1,2], Zoltán Németh[3], Veronika Varga[3], Tamás Weidinger[4], Ville Leinonen[1],
Taina Yli-Juuti[1], and Imre Salma[3]
[1] Department of Applied Physics, University of Eastern Finland, P.O. Box 1627, 70211 Kuopio, Finland
[2] Department of Environmental and Biological Sciences, University of Eastern Finland, P.O. Box 1627,
70211 Kuopio, Finland
[3] Institute of Chemistry, Eötvös University, H-1518 Budapest, P.O. Box 32, Hungary
[4] Department of Meteorology, Eötvös University, H-1518 Budapest, P.O. Box 32, Hungary
*Correspondence to*: Imre Salma (salma@chem.elte.hu) and Santtu Mikkonen (santtu.mikkonen@uef.fi)
**Abstract.** Multiple atmospheric properties were measured semi-continuously in the Budapest platform
for Aerosol Research and Training Laboratory for a time interval of 2008-2018. Dataset of 6 full
measurement years during a decennial time interval were subjected to statistical time trend analyses by
an advanced dynamic linear model and a generalized linear mixed model. The main interest in the
analysed data set was on particle number concentrations in the diameter ranges from 6 to 1000 nm ($N_{6-1000}$),
from 6 to 100 nm ($N_{6-100}$, ultrafine particles), from 25 to 100 nm ($N_{25-100}$) and from 100 to 1000 nm
($N_{100-1000}$). These data were supported by concentrations of $SO_2$, CO, NO, $NO_x$, $O_3$, $PM_{10}$ mass, air
temperature, relative humidity, wind speed, atmospheric pressure, global solar radiation, condensation
sink, gas-phase $H_2SO_4$ proxy, classes of new aerosol particle formation (NPF) and growth events and
meteorological macro-circulation patterns. The trend of the particle number concentrations derived as a
change in the statistical properties of background state of the data set decreased in all size fractions over
the years. Most particle number concentrations showed decreasing decennial statistical trends. The
estimated annual mean decline of $N_{6-1000}$ was (4–5)% during the 10-year measurement interval, which
corresponds to a mean absolute change of $-590$ cm$^{-3}$ in a year. This was interpreted as a consequence
of the decreased anthropogenic emissions mainly from road traffic. Similar trends were not observed
for the air pollutant gases. Diurnal statistical patterns of particle number concentrations showed
tendentious variations, which were associated with typical diurnal activity–time pattern of inhabitants
in cities, particularly of vehicular road traffic. The trend patterns for NPF event days contained a huge
peak from late morning to late afternoon, which is unambiguously caused by NPF and growth processes.
These peaks were rather similar to each other in the position, shape and area on workdays and holidays,
which implies that the dynamic and timing properties of NPF events are not substantially influenced by
anthropogenic activities in central Budapest. Diurnal pattern for $N_{25-100}$ exhibited the largest relative
changes, which were related to particle emissions from high-temperature sources. The diurnal pattern
for $N_{100-1000}$ – which represents chemically and physically aged particles of larger spatial scale – were
different from the diurnal patterns for the other size fractions.



## 1 Introduction

Atmospheric aerosol can be characterised by various properties. There are several important phenomena and processes in which individual particles play a role. In these cases, particle number concentrations or particle number size distributions are the relevant metrics. Number concentrations of (insoluble) particles produce adverse effects on human health (Oberdörster et al., 2005; Rich et al., 2012; Cassee et al., 2013; Braakhuis et al., 2014; Ostro et al., 2015; Schmid and Stoeger, 2016; Ohlwein et al., 2019). Individual particles and their properties are also important in cloud formation processes and, therefore, in indirect aerosol climate forcing (Makkonen et al., 2009; Merikanto et al, 2009; Sihto et al., 2011; Kerminen et al., 2012; Carslaw et al., 2013; Gordon et al., 2016). Particle numbers and associated size distributions are relevant properties in several optical interactions in the atmosphere (e.g. Moosmuller et al., 2009) and in various surface-controlled chemical reactions (e.g. Pöschl et al., 2007).

In the global troposphere, it is the new aerosol particle formation (NPF) and consecutive growth process that is the dominant source of particle numbers (Spracklen et al., 2006; Yu et al., 2010; Kulmala et al., 2013; Dunne et al., 2016). This source type occurs in various atmospheric environments around the world and produces secondary particles (Kerminen et al., 2018 and references therein). The major anthropogenic source of (primary) particles is combustion. It includes traffic exhaust mainly from diesel engines, fuel or waste burning in industrial and domestic installations, residential heating and cooking (Paasonen et al., 2016; Masiol et al., 2018). Nanotechnology and its products can have importance in some limited or occupational environments. In large cities and in longer time intervals, primary particles often prevail over secondary particles (Brines et al., 2015; Salma et al., 2017; Saha et al., 2018).

Ultrafine (UF) particles (with a diameter $d<100$ nm) account for most of the particle number concentrations but have usually negligible contribution to particulate matter (PM) mass. This implies that particle numbers are not covered by legislative regulations on the ambient air quality, which are ordinary based on the PM mass. Particle number concentrations have not been promulgated among the air quality standards yet. There are, however, mitigation policies and control regulations, which intend to reduce their ambient levels as part of an overall air-quality improvement strategy since 1990s. The legislations, for instance in the EU including Hungary, focus on the particle emissions from diesel engines (Giechaskiel et al., 2018). There were some important changes in the car emissions during the time interval under the investigation in this study. These included the introduction of Euro 5 and 6 regulations for light-duty vehicles in January 2011 and Euro VI regulations for heavy-duty vehicles in September 2015 (the number of emitted particles with diameters >23 nm should be <6×10$^{11}$ km$^{-1}$ for type approval). A prerequisite for the efficient operation of exhaust after treatment devices is having fuel with low sulfur content. The reduction of sulfur in diesel fuel for on-road transport was decreased after several previous phases to <10 ppm in January 2009 (Directive 2009/30/EC). Sulfur content in fuels for mobile non-road diesel vehicles – including mobile machinery, agricultural and forestry





tractors, inland waterway vessels and recreational crafts – was limited at a level of 1000 ppm from 2008
and at 10 ppm from 2011. The unsuitable/dangerous fuel types for domestic heating are also listed, their
emission factors are determined, and the accumulated information is disseminated among potential
users. As far as secondary particles are concerned, it is not straightforward to reduce their concentration
levels because the effects of gaseous and aerosol species on the NPF are complex and uncertain due to
nonlinear relationship and feedbacks in their related processes.
It is relevant to investigate the potential changes, namely overall and diurnal tendencies of particle
number concentrations from different sources on longer run because of their role in both health-risk and
climate-change issues. The major source types of particle numbers can be separated by measuring their
size distributions. Atmospheric NPF events produce particles of the nucleation mode, which occurs
intermittently, and which gradually merges into the larger Aitken mode. High temperature emission
sources ordinarily produce Aitken-mode particles, while transformation processes (physical and
chemical aging) of existing particles in the atmosphere give rise to the accumulation mode. An important
property of the nucleation- and Aitken-mode particles is that their residence time is limited to several
hours (Raes et al., 2000; Salma et al., 2011). This is different from accumulation-mode particles, which
reside in the air up to 7 days. This means that the particles of the former two modes are present in the
air until their sources are active, and that their concentrations can change substantially and rapidly over
a day (e.g. Mikkonen et al., 2011a, Salma et al., 2014, 2017; Paasonen et al., 2016). This is advantageous
when source types are to be identified or quantified. At the same time, the relatively short residence time
is not beneficial when time trends are to be studied and derived. The limited residence time can cause
additional, substantial and sudden variability in time with or without time patterns, which can complicate
the evaluation.
Particle number concentrations or particle number size distributions in the relevant diameter range (i.e.
from few nanometers to ca. 1 μm) are measured for various purposes. They include fundamental studies
on atmospheric nucleation and particle growth phenomena, which usually require semi-continuous long-
term measurements. The related experimental data sets have been accumulating gradually (Wehner and
Wiedensohler, 2003; Asmi et al., 2013; Kerminen et al., 2018; Nieminen et al., 2018). They can also be
exploited for time trend analysis by using appropriate statistical models. At present, however, knowledge
on time trends particularly in various size fractions and over several years is largely lacking with few
recent exceptions (Masiol et al., 2018; Saha et al., 2018; Sun et al., 2019).
Research activities dedicated to NPF and growth events in Budapest have been going on since November
2008. Measurements for 6 full years were realised in the city centre at a single fixed location. Semi-
continuous and critically evaluated data sets consisting of particle number size distributions,
concentrations of criteria air pollutants and meteorological data were available for the study. They were
combined in a coherent set, which was utilised in two statistical models developed specifically to
determine the time trends for particle number concentrations in several important size fractions from
2008 to 2018. The main objectives of this study are to present and discuss the statistical models, to
interpret their results on time trends and diurnal variability, to quantify the change rates, and to relate
the temporal tendencies to different atmospheric sources, processes and environmental circumstances.
**2 Methods**
**2.1 Measurements**
Most experimental data dealt with in the present study were obtained at a single urban site, namely at
the Budapest platform for Aerosol Research and Training (BpART) research laboratory (N 47° 28' 29.9",
E 19° 3' 44.6", 115 m above mean sea level). This location represents a well-mixed, average atmospheric
environment for the city centre of Budapest due to its geographical and meteorological conditions
(Salma et al., 2016a). The local emissions include diffuse urban traffic exhaust, household/residential
emissions and limited industrial sources together with some off-road transport (diesel rail, shipping and
airplane emissions). Experimental data for 6 full-year-long time intervals, i.e. from 3 November 2008
to 2 November 2009, from 13 November 2013 to 12 November 2014, from 13 November 2014 to 12
November 2015, from 13 November 2015 to 12 November 2016, from 28 January 2017 to 27 January
2018 and from 28 January 2018 to 27 January 2019 were available for this single site. A decennial time
interval from 03 November 2008 to 02 November 2018 was considered in the statistical analysis. Local
time (LT=UTC+1 or daylight-saving time, UTC+2) was chosen as the time base of the data processing
because the ordinary daily activities of inhabitants substantially influence the atmospheric
concentrations and several processes in cities (Salma et al., 2014).
The major aerosol measuring system was a flow-switching type differential mobility particle sizer
(DMPS, Alto et al., 2001). It records particle number concentrations in an electrical mobility diameter
range from 6 to 1000 nm in the dry state of particles (with a relative humidity of RH<30%) in 30
channels (Salma et al., 2011). The measuring system was updated twice; in spring 2013 and winter 2016.
Its major parts including a differential mobility analyser (DMA, Hauke-type with a length of 28 cm) and
a condensation particle counter (CPC, TSI model 3775) remained, however, unchanged. They were
cleaned and serviced. The diameter resolution of the DMA was also calibrated during the updates.
Several data validation or comparative exercises were realised over the years; the most extensive inter-
comparison was realised in summer 2015 (Salma et al., 2016a) and autumn 2019. The time resolution
of the DMPS measurements was approximately 10 min in the year 2008–2009 and it was 8 min from 13
November 2013 on. The sampling inlet was installed at a height of 12.5 m above the street level. There
was no upper-size cut-off inlet applied to the sampling line, and a rain shield and insect net were only
adopted. The measurements were performed according to the international technical standard
(Wiedensohler et al., 2012).




Meteorological data for air temperature ($T$), relative humidity (RH), wind speed (WS), wind direction
and atmospheric pressure ($p$) were obtained from a measurement station of the Hungarian
Meteorological Service (HMS) operated in a distance of ca. 70 m from the BpART laboratory by
standardised methods (Vaisala HMP45D humidity and temperature probe, Vaisala WAV15A
anemometer, Vaisala pressure, all Finland) with a time resolution of 10 min. Global solar radiation
(GRad) data were measured by a CMP11 pyranometer (Kipp and Zonnen, The Netherlands) at another
station of the HMS situated in 10 km in Eastern direction with a time resolution of 1 h. Concentrations
of pollutants $SO_2$, CO, NO, $NO_x$, $O_3$, and $PM_{10}$ mass were acquired from a measurement station of the
National Air Quality Network in Budapest in Széna Square, which is located in the upwind prevailing
wind direction in a distance of 4.5 km from the BpART laboratory. They are measured by UV
fluorescence (Ysselbach 43C), IR absorption (Ysselbach 48C), chemiluminescence (Thermo 42C), UV
absorption (Ysselbach 49C) and beta-ray attenuation (Thermo 5014I) methods, respectively with a time
resolution of 1 h.
The availability of the DMPS data over the six one-year-long time intervals were 95, 99, 95, 73, 99 and
90%, respectively. The meteorological data were accessible in >90% of time in each year, while the
concentration data for key pollutants were available in >85% of the yearly time intervals.
**2.2 Data treatment**
Particle number concentrations in the diameter ranges 1) from 6 to 1000 nm ($N_{6–1000}$), 2) from 6 to 100
nm ($N_{6–100}$), 3) from 25 to 100 nm ($N_{25–100}$) and 4) from 100 to 1000 nm ($N_{100–1000}$) were calculated from
the measured and inverted DMPS data. The size ranges were selected to represent 1) the total particles,
2) UF particles, 3) UF particles emitted mainly from incomplete combustion (and partially grown by
condensation; this size ranges is dominated by primary particles in cities in most of the time) and 4)
physically and chemically aged particles which usually represent larger spatial extent, respectively
(Salma et al., 2014, 2017).
Condensation sink (CS) for vapour molecules onto the surface of existing aerosol particles was
calculated for discrete size distributions (Kulmala et al., 2001, 2012; Dal Maso et al., 2002, 2005). Dry
particle diameters were considered in the calculations and condensing vapour was assumed to have
sulphuric acid properties.
One of the key components for NPF events is the gas-phase $H_2SO_4$ (Sipilä et al., 2010; Sihto et al.,
2011). It is challenging to measure its atmospheric concentration and, therefore, the experimental data
for long time intervals are rare. The relative effects of gas-phase $H_2SO_4$ are, however, often estimated
by deriving its proxy value. In this study, the $H_2SO_4$ proxy was calculated according to Mikkonen et al.
(2011b), where the best proxy was based on GRad, $SO_2$ concentration, RH and CS. The proxy is defined
for GRad>10 W $m^{-2}$. Other widely used proxy was introduced by Petäjä et al. (2009), but that was


created for clean boreal forest environment. The most recent proxy from Dada et al. (2020) is currently
under review and has not been tested against the proxy used here. All experimental data were used with
their maximum time resolution.
The influence of large-scale weather types was considered on a daily basis by including codes for macro-
circulation patterns (MCPs) invented specifically for the Carpathian Basin (Péczely, 1957; Károssy,
2016). The classification is based on the extension and development of cyclones and anticyclones
relative to the Carpathian Basin via the daily sea-level pressure maps constructed for 00:00 UTC in the
North-Atlantic–European region. Basic information on the MCPs are summarised in Table 1.
**Table 1.** Macro-circulation patterns (Péczely codes) and their seasonal and annual occurrences in the Carpathian
Basin for years 1958–2010 (Maheras et al., 2018).

| No. | Code | Description | Occurrence (%) | | | | |
|-----|------|-------------|--------|--------|--------|--------|--------|
| | | | Winter | Spring | Summer | Autumn | Annual |
| 1 | mCc | Cyclone with a cold front over NE Europe, N wind | 7.3 | 11.3 | 12.1 | 8.0 | 9.7 |
| 2 | AB | Anticyclone over the British Isles, N wind | 5.6 | 7.1 | 8.6 | 6.4 | 6.9 |
| 3 | CMc | Mediterranean cyclone with a cold front over S Europe, N wind | 2.5 | 3.5 | 1.8 | 1.9 | 2.4 |
| 4 | mCw | Mediterranean cyclone with a warm front over NE Europe, S wind | 9.2 | 9.7 | 5.7 | 7.2 | 7.9 |
| 5 | Ae | Anticyclone over E Europe, S wind | 14.2 | 11.3 | 7.3 | 17.6 | 12.6 |
| 6 | CMw | Mediterranean cyclone with a warm front over S Europe, S wind | 8.9 | 8.7 | 3.7 | 8.3 | 7.4 |
| 7 | zC | Highly developed cyclone over N Europe, W wind | 5.0 | 3.2 | 2.7 | 2.9 | 3.5 |
| 8 | Aw | Anticyclone over W Europe, W wind | 13.1 | 11.2 | 20.8 | 12.8 | 14.6 |
| 9 | As | Anticyclone over S Europe, W wind | 7.0 | 4.4 | 2.9 | 5.6 | 4.9 |
| 10 | An | Anticyclone over N Europe, E wind | 10.9 | 12.8 | 11.3 | 10.1 | 11.3 |
| 11 | AF | Anticyclone over Fennoscandia, E wind | 2.8 | 5.2 | 5.9 | 3.7 | 4.4 |
| 12 | A | Anticyclone over the Carpathian Basin, changing wind direction | 11.8 | 7.3 | 13.3 | 13.3 | 11.4 |
| 13 | C | Cyclone over the Carpathian Basin, changing wind direction | 1.7 | 4.3 | 3.9 | 2.2 | 3.0 |


Each data line containing the date and time, concentrations, CS, $H_2SO_4$ proxy, meteorological data and
MCP codes was further labelled by several indices on a daily basis. These labels served to differentiate
between various environmental conditions, which can lead to substantial changes in some variables
(Salma et al., 2014). The workdays were marked by label WD, while the holidays were denoted by label
HD. Varying classes of NPF event days were also labelled differently. The classification was





accomplished via the particle number size distribution surface plots (Dal Maso et al., 2005; refined in
Németh et al., 2018 for urban sites) on a daily basis. The main classes were: NPF event days (marked
by label NPF), non-event days (label NE), days with undefined character and days with missing data.
The earliest estimated time of the beginning of a nucleation ($t_1$) was also derived (Németh and Salma,
2014) and was added to the data record as a parameter. Finally, the data lines were labelled according
to the actual technical status of the DMPS system. The data obtained from the beginning of the
measurements to the 1[st] update was labelled as S1, the data derived between the 1[st] and 2[nd] updates were
label as S2, and label S3 was given to the data obtained after the 2[nd] update.

**2.3 Statistical modelling**

Atmospheric data are usually not normally distributed, and, therefore, non-parametric methods are often
used to detect their long-term trends (Asmi et al., 2013; Masiol et al., 2018). The coherent data set
prepared as described in Sect. 2.2 was analysed in two ways. First, time trends for concentrations of
particles and air pollutants were estimated by using a dynamic linear model (DLM) method. Secondly,
the factors affecting the changes in particle concentrations were detected with a generalized linear mixed
model (GLMM).

**2.3.1 Dynamic linear model**

Dynamic linear models (Durbin and Koopman, 2012; Petris et al., 2009; Laine, 2020) are state-of-the-
art tools for time trend detection. The trend is seen as a statistical change in the properties of the
background state of the system. Although changes in aerosol concentrations have previously been
approximated with linear trends (e.g. Sun et al., 2019), this is not always the most suitable method since
the processes affecting the concentrations are continuously evolving over time. Additionally, time series
of atmospheric measurements can include multiple time-dependent cycles (e.g. seasonal and diurnal
cycles) which are typically non-stationary – meaning that their distributional properties change over
time. The DLM approach effectively decomposes the data series into basic components such as level,
trend, seasonality and effect of external forcing by describing statistically the underlying structure of the
process that generated the measured data. All these components are defined by Gaussian distributions,
and they are allowed to vary in time, and the significance and magnitude of this variation can also be
modelled and estimated. In the basic setup of DLM, the sign or the magnitude of the trend is not defined
in advance by the model formulation but estimated from the data. The method can detect and quantify
trends, but the explanations for the observed changes is provided by the user. Nevertheless, it determines
if the observations are consistent with the selected model. We used the DLM to explain variability in
the particle concentration time series using following components: locally linear mean level, trend,
seasonal effect, autoregressive component and noise. The evolution of the investigated concentrations –
after the seasonal and noise components were filtered out – is modelled by using the smoothed mean



level. Here, the change in the mean level is the trend of the variable. The statistical model can be
described by the following equations (Mikkonen et al., 2015):
$y_t = \mu_t + \gamma_t + \eta_t + \varepsilon_{obs}$ , $\varepsilon_{obs} \sim N(0, \sigma_t^2)$,                                        (1)
$\mu_t = \mu_{t-1} + \alpha_t + \varepsilon_{level}$, $\varepsilon_{level} \sim N(0, \sigma_{level}^2)$,                                        (2)
$\alpha_t = \alpha_{t-1} + \varepsilon_{trend}$, $\varepsilon_{trend} \sim N(0, \sigma_{trend}^2)$,                                        (3)
where $y_t$ is the investigated concentration at time $t$, $\mu_t$ is the mean level and $\alpha_t$ is the change in the level
from time $t$–1 to time $t$, $\gamma_t$ is the seasonal component and $\eta_t$ is an autoregressive error component. Here,
this latter level is fixed. The Gaussian stochastic $\varepsilon$ terms are used for the observation uncertainty and
for random dynamics of the level and the trend. The seasonal component $\gamma_t$ contains dummy variables
for each month, so it has a different value for each month with a condition that 12 consecutive months
sum to zero. More detailed description on how the model is written through state space equation can be
found in Mikkonen et al. (2015).
**2.3.2 Generalized linear mixed model**
Linear mixed models (McCulloch et al., 2008) belong to the family of models that combine several
different kinds of models used in multivariate analysis when the data do not fulfil the standard
independency and homogeneity assumptions. This is the normal case with atmospheric and
climatological measured variables (e.g. Mikkonen et al., 2011a). The main goal of the mixed models is
to estimate not only the mean of the measured response variable but also the variance-covariance
structure of the data, which makes the model more valid for complex atmospheric data. In addition,
modelling the (co)variances of the variables reduces the bias of the estimates, and prevents
autocorrelation of the residuals. The model is constructed from general linear model, written in matrix
format as $y=X\beta+\varepsilon$, by adding a so-called random component (denoted $Zu$) to the model, thus the model
is given by $y= X\beta+Zu+\varepsilon$. Here, if we let $n$ equal to number of observations, $p$ equal to number of fixed
parameters and $q$ equal to number of random parameters in the model, $y$ is the ($n\times1$) vector of
measurements of the variable of interest, $\beta$ denotes the unknown ($p\times1$) vector of intercept and slope
estimates of the model, $X$ is the ($n\times p$) matrix of observations from predictor variables and $\varepsilon$ contains
the residuals of the model. In the random part, $Z$ is the ($n\times q$) design matrix for the ($q\times1$) vector of
random covariates $u$ with a $q$-dimensional normal distribution. With adequate choices of the matrix $Z$,
different covariance structures $Cov(u)=G$ and $Cov(\varepsilon)=R$ can be defined and fitted. Successful modelling
of variances and covariances of the observations provides valid statistical inference for the fixed effects
$\beta$ of the mixed model. In contrast to general linear models, the error terms $\varepsilon$ can be correlated, which
makes the modelling more robust. It follows from this that the distribution of observations can be
described by a normal distribution with the expectation of $\bar{X}$ and covariance matrix $V$, which is given
by $V=ZGZ'+R$. With GLMM, it is possible to reliably detect the factors which affect particle number



concentrations or which act as indicators for their different sources. The model can be expressed in a
mathematical form as (Mikkonen et al., 2011a):
$$N_{Di} = (\beta_0 + \beta_{setup} + u_m) + \alpha_d + (\beta_{wd} \cdot \beta_E) \cdot X_{Ti} + (\beta_1 + v_{1m}) \cdot SO_{2,i} + (\beta_2 + v_{2m}) \cdot NO_{2,i} +$$
$$(\beta_3 + v_{3m}) \cdot O_{3,i} + \beta_4 \cdot GRad_i + \beta_5 \cdot RH_i + \beta_6 \cdot MCP_i, \tag{4}$$
where $N_{Di}$ is the number concentration in selected size range in time $i$, $\beta_0$ is a model intercept, $\beta_{setup}$ is
a correction term for changes in the measurement system due to two major upgrades, $u_m$ is vector of
random intercepts different for each month, $\alpha_d$ is average change of $N_{Di}$ per day (i.e. slope of trend),
$\beta_{wd}$ and $\beta_E$ are coefficients for workday and NPF event day, respectively, and $X_{Ti}$ is the corresponding
vector showing the type of the day (in both means: WD/HD and E/NE) in time $i$, $\beta_1 - \beta_5$ are fixed
coefficients for $SO_2$, $NO_2$, $O_3$, $GRad$ and $RH$, respectively, $\beta_6$ is the vector of coefficients for different
macro-circular patterns (MCP) and $v_m$ are the random, month specific slopes for $SO_2$, $NO_2$, $O_3$ and
$GRad$. The coefficients of the model can be interpreted in a similar manner as multivariate regression
or general linear models, just with an addition of month-specific effects for given variables.
**3 Results and discussion**
An overview on annual averages of the data analysed in this study is given in Table 2. Annual insolation,
which expresses the total energy density at the receptor site, was derived from the individual $GRad_i$ data
as $Q = n_d \times 24 \times 3600 \times \sum_i GRad_i$ over the year of interest ($n_d$ is the number of day in the year). Since the
major sources of particles in cities include road vehicles and atmospheric nucleation, we added some
indicative data on these specific sources as well. The median particle number concentrations are
basically in line with many other comparable cities in the world (e.g. Kerminen et al., 2018; Masiol et
al., 2018). They indicate a decreasing change (except for $N_{100–1000}$) over the years 2008–2018. At the
same time, the annual averages of the other concentrations, meteorological data and auxiliary variables
did not change substantially. Annual mean relative occurrence frequency of NPF events stayed almost
constant with a mean and SD of (20±4)%, except for the measurement year 2015–2016 when it was
unusually small. It is worth adding that the NPF increases the existing particle number concentrations
in Budapest by a factor of approximately 2 on event days (Salma et al., 2017). The annual medians for
the particle formation rate and particle growth rate also stayed constant and seemingly varied only as
fluctuations within ca. ±20% and ±8%, respectively. The number of passenger cars was registered in
Budapest remained constant within ±5%, while the share of the diesel-powered passenger cars increased
modestly by a rate of approximately 12% from 2008 to 2018 (KSH, 2019). The number (ca. 4000) of
buses registered in Budapest and the share (98%) of the diesel-power buses on the national bus fleet
remained constant.





**Table 2.** Annual medians of particle number concentrations in the diameter ranges from 6 to 1000 nm ($N_{6-1000}$),
from 6 to 100 nm ($N_{6-100}$), from 25 to 100 nm ($N_{25-100}$) and from 100 to 1000 nm ($N_{100-1000}$), concentrations of $SO_2$,
CO, NO, $NO_x$, $O_3$, $PM_{10}$ mass, annual means of air temperature ($T$), relative humidity (RH), wind speed (WS),
atmospheric pressure ($P$) and annual insolation ($Q$), annual mean relative occurrence frequency of nucleation
($f_{NPF}$), annual median formation rate of particles with a diameter of 6 nm ($J_6$), annual median growth rate of
particles with a diameter of 10 nm ($GR_{10}$; for the rates, see Salma and Németh, 2019), number of passenger cars
registered in Budapest (Cars), the mean age and the share of diesel-powered vehicles (Diesel) separately for the 1-
year-long measurement time intervals.

| Variable | Unit | 2008–2009 | 2013–2014 | 2014–2015 | 2015–2016 | 2017–2018 | 2018–2019 |
|---|---|---|---|---|---|---|---|
| $N_{6-1000}$ | $10^3$ cm$^{-3}$ | 11.5 | 9.7 | 9.3 | 7.5 | 8.6 | 8.3 |
| $N_{6-100}$ | $10^3$ cm$^{-3}$ | 9.1 | 7.2 | 6.9 | 5.7 | 6.8 | 6.5 |
| $N_{25-100}$ | $10^3$ cm$^{-3}$ | 5.1 | 4.3 | 4.1 | 3.3 | 3.6 | 3.2 |
| $N_{100-1000}$ | $10^3$ cm$^{-3}$ | 1.79 | 2.2 | 2.0 | 1.56 | 1.49 | 1.53 |
| $SO_2$ | µg m$^{-3}$ | 5.0 | 4.8 | 4.6 | 4.8 | 4.5 | 5.2 |
| CO | µg m$^{-3}$ | 547 | 488 | 577 | 513 | 534 | 624 |
| NO | µg m$^{-3}$ | 13.3 | 19.2 | 23 | 17.6 | 20 | 17.0 |
| $NO_x$ | µg m$^{-3}$ | 58 | 80 | 89 | 72 | 79 | 73 |
| $O_3$ | µg m$^{-3}$ | 23 | 14.8 | 19.6 | 25 | 20 | 21 |
| $PM_{10}$ | µg m$^{-3}$ | 33 | 31 | 39 | 29 | 28 | 36 |
| $T$ | °C | 12.0 | 13.2 | 13.2 | 12.9 | 13.2 | 13.3 |
| RH | % | 64 | 69 | 64 | 69 | 63 | 67 |
| WS | m s$^{-1}$ | 2.5 | 2.6 | 2.8 | 2.7 | 2.9 | 2.5 |
| $P$ | hPa | 1001 | 1003 | 1005 | 1004 | 1004 | 1004 |
| $Q$ | GJ m$^{-2}$ y$^{-1}$ | 4.75 | 4.51 | 4.71 | 4.67 | 4.95 | 4.81 |
| $f_{NPF}$ | % | 24 | 20 | 23 | 13.0 | 23 | 20 |
| $J_6$ | cm$^{-3}$ s$^{-1}$ | 4.2 | 3.5 | 4.4 | 4.6 | 6.3 | 5.3 |
| $GR_{10}$ | nm h$^{-1}$ | 7.6 | 6.6 | 6.5 | 8.0 | 7.5 | 7.5 |
| Cars[*] | $10^3$ pcs | 582 | 573 | 584 | 597 | 634 | 659 |
| Age[*] | y | 10.8 | 13.0 | 13.4 | 13.7 | 14.1 | 14.2 |
| Diesel[*] | % | 20 | 24 | 26 | 28 | 29 | n.a. |

[*] Status at the end of years 2009, 2013, 2014, 2015, 2017 and 2018, respectively.
n.a.: not yet available.
**3.1 Decennial time scale**
Overall statistical time trends for particle number concentrations in various size fractions obtained by
the DLM are displayed in Figure 1. The curves confirm that the $N_{6-1000}$, $N_{6-100}$ and $N_{25-100}$ indeed
decreased in Budapest between 2008 and 2018, while the change in $N_{100-1000}$ was not significant. The
decline mostly took place in a monotonical manner except for perhaps the interval of summer 2016–
spring 2017, when some partial/local increase could be realised for $N_{6-1000}$ and $N_{6-100}$.



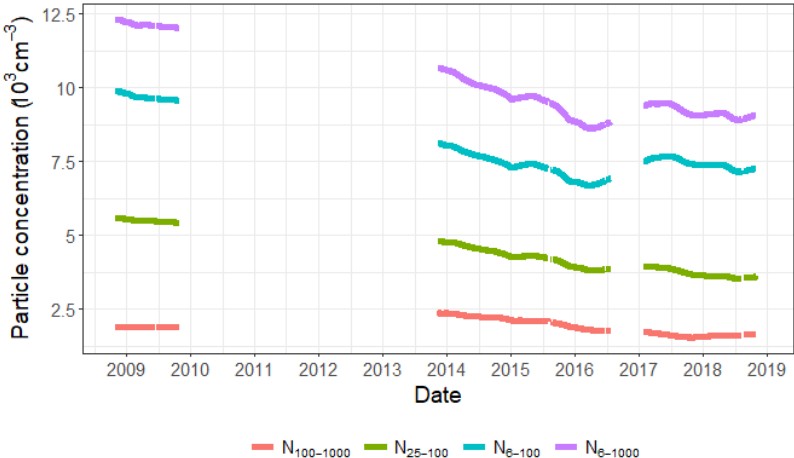

**Figure 1.** Statistical time trends of particle number concentrations in the diameter ranges from 6 to 1000 nm ($N_{6-1000}$), from 6 to 100 nm ($N_{6-100}$), from 25 to 100 nm ($N_{25-100}$) and from 100 to 1000 nm ($N_{100-1000}$) derived by DLM over a decennial interval.

There are several important sources, sinks and atmospheric transformation processes including environmental conditions which can influence the atmospheric concentrations. The major sources include both high-temperature emissions and NPF events as discussed in Sect. 1. The latter source is affected by concentrations of precursor and other trace gases, meteorological properties for photochemical reactions, and the interactions among gas-phase chemical species of different origin/type with respect the formation yield of condensing vapours (Kulmala et al., 2014; McFiggans et al., 2019). The air pollutants listed in Table 2 and gas-phase $H_2SO_4$ proxy – which are known or expected to affect particle number concentrations – did not exhibit decreasing statistical trend between 2008 and 2018 (Fig. 2). On one hand, this decoupling confirms that the causes of the decrease in particle number concentrations are not primarily related to meteorological conditions because they would jointly affect the gas concentrations as well (if their sources are more-or-less constant over a certain time interval). On the other hand, the constant gas concentrations suggest that the decreasing trend in particles does not seem to be related to the major precursors or interacting gaseous chemical species (such as $SO_2$, $H_2SO_4$ or $NO_2$).



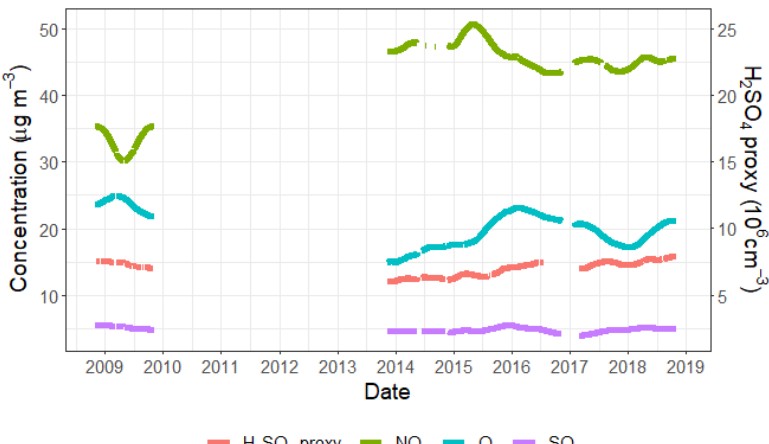

**Figure 2.** Statistical time trends of gas-phase $H_2SO_4$ proxy, $SO_2$, $O_3$ and $NO_2$ derived by DLM over the decennial interval.

As far as the meteorological conditions are concerned, some of them such as WS, boundary mixing layer height and $T$ have previously been shown to influence the temporal variation of aerosol particles (e.g. Birmili et al., 2001; Mikkonen et al., 2011a). The annual means of possibly relevant properties and parameters in Table 2 – except for the particle number concentrations (which are under the investigation) and the fraction of diesel cars – did not show any obvious dependency; they virtually stayed constant over the years of interest. The possible effect of different weather conditions on the concentrations are studied separately by the GLMM and are discussed in Sect. 3.2.2. There were also no substantial and extensive urban constructions in the area (which could influence the urban air flow) nor larger systematic changes in the traffic circulation around the sampling site in the time interval considered. Therefore, the decline in the particle number concentrations is most likely interpreted as a consequence of the decreased anthropogenic particulate emissions in Budapest. The related source sectors can include vehicular road traffic and household heating/cooking. The decline happened at an increasing share of the diesel passenger cars and straitened emission control on (diesel) vehicles.

The average decrease rates of particle number concentrations as derived from both the DLM and GLMM statistical approaches are summarised in Table 3. The rates are shown as obtained from the models and scaled for the 10-year measurement interval to ensure the comparability of the slopes. The relative mean changes in % per year were expressed with respect to the starting value (mean of the first year). There are some differences between the corresponding results of the two models, which were caused by standardising the concentrations with the predictors in the models and by handling the upgrades of measurement setup differently. The changes in all size fractions were on the same level and only minor differences could be seen. As the estimates always contain some uncertainty, these differences are not considered as statistically significant. The largest difference between the two models was observed for





$N_{100-1000}$ (which had the lowest absolute concentrations). One possible cause for this might be that
GLMM standardises the results for variables indicating anthropogenic emissions and thus the size
fraction that is the most sensitive for the emissions has the strongest effect. Considering all these, the
rates from the two statistical models agree well. Furthermore, the rates for $N_{6-1000}$ and $N_{6-100}$ were
identical. This is explained by the fact that these two size fractions are strongly connected; the typical
$N_{6-100}/N_{6-1000}$ mean ratio in central Budapest is 75–80% (Salma and Németh, 2019). Small difference
was also seen for $N_{25-100}$. In urban areas, this size fraction is mainly composed of particles from high-
temperature emission sources. The source types responsible for the observed decline are further
discussed in Sect. 3.2.1.
**Table 3.** Decrease rates of particle number concentrations in the diameter ranges from 6 to 1000 nm, from 6 to
100 nm, from 25 to 100 nm and from 100 to 1000 nm obtained by the dynamic linear model and generalized linear
mixed model as a mean absolute change per year during the 10-year measurement interval and as a relative mean
change per year with respect to the mean value of the first year.

| Size fraction (nm) | Dynamic linear model | | Generalized linear mixed model | |
|---|---|---|---|---|
| | Mean change/year (cm$^{-3}$) | Relative mean change (%/year) | Mean change/year (cm$^{-3}$) | Relative mean change (%/year) |
| 6–1000 | –510 | –4 | –660 | –5 |
| 6–100 | –400 | –4 | –480 | –5 |
| 25–100 | –310 | –6 | –360 | –5 |
| 100–1000 | –50 | –3 | –180 | –8 |


Our results concerning the decennial change rates (and our conclusions with regard to their causes
mainly discussed in Sect. 3.2.1) are comparable and are in line with some other very recent studies. Sun
et al. (2019) investigated the statistical concentration trends in particle numbers (and equivalent black
carbon mass) at multiple urban, rural or background sites within the German Ultrafine Aerosol Network.
Decreasing annual slopes of –(7.0–1.7)% were obtained for several size fractions (which are different
from our intervals), and the most likely factors for the decreasing trends were assigned to declining
anthropogenic emissions due to emission mitigation policies of the EU. Masiol et al. (2018) evaluated
statistical time trends of particle number concentrations in various size fractions (which are different
again from the previous and present studies) in Rochester, NY, USA, and obtained a typical decline rate
of –4.6% per year for total particles. These outcomes and our data as well seem to be different from the
results obtained by Saha et al. (2018) in the urban Pittsburgh, PA, USA by comparing two intervals of
2001–2002 and 2016–2017. It should be mentioned that in the latter research, the experimental setup
for measuring particle number size distributions had a lower diameter limit of detection at 11 nm, some
methodological approaches (e.g. classification of events) were different from ours and that the time trend
was not derived by statistical modelling. The authors concluded that both the frequency of NPF events
and their dynamic properties were reduced by (40−50)% over the past 15 years, resulting in ca. 48%



reduction of UF concentrations. The changes were attributed to dramatic reductions in SO$_2$ emissions in
the larger region.

**3.2 Diurnal time scale**

Diurnal statistical patterns of the particle number concentrations in different size fractions were
predicted by the GLMM considering the following variables: GRad, RH, concentrations of SO$_2$, NO$_2$,
O$_3$, and labels for workdays/holidays, for NPF event days/non-event days and for MCP codes. The initial
screening for possible prediction variables was done in earlier papers. Studies such as Hyvönen et al.
(2005), Mikkonen et al. (2006) and Nieminen et al. (2014) suggested that meteorological and trace gas
variables affect NPF. Furthermore, e.g. Mikkonen et al. (2011a), Guo et al. (2012) and Zaidan et al.
(2018) studied the factors which influence the growth of freshly formed particles as well as the
concentrations of particles in larger size fractions and specified the possible predictors. All variables
found in these screenings and measured at our site were tested one-by-one in the GLMM model in a
stepwise manner. In each step, the significance of the added or removed variable was investigated by a
likelihood ratio test (e.g. Pinheiro and Bates, 2000) until the final model shown in Eq. 4 was formed.
The effect of the H$_2$SO$_4$ proxy was also tested, and the results for the daytime concentrations were
similar to those obtained with the selection of variables above. The modelling results for night-time
were, however, biased since the proxy is defined for GRad>10 W m$^{-2}$, and, therefore, we decided not to
include the proxy into the final model.

**3.2.1 Diurnal statistical patterns**

Modelled diurnal pattern of particle number concentrations for event days on workdays, event days on
holidays, non-event days on workdays and non-event days on holidays separately for different size
fractions are shown in Fig. 3. The curves on Fig. 3a–c resemble tendentious variations, which can be
associated with typical diurnal activity–time pattern of inhabitants in cities, particularly with road traffic.
They are also perfectly in line with the mean diurnal tendencies of experimentally determined
concentrations in central Budapest (Salma et al., 2014; 2017) and are consistent with the time variations
in many other European cities (Hussein et al., 2004; Aalto et al., 2005; Moore et al., 2007; Avino et al.,
2011; Dall'Osto et al., 2013).
In the statistical diurnal patterns of UF particles (Fig. 3b), there is a huge peak from late morning to late
afternoon on event days. This is unambiguously caused by NPF and growth process. The peaks on
workdays and holidays are rather similar to each other in the position, shape and magnitude (area), which
means that the dynamics and timing of NPF events in general are not substantially influenced by
anthropogenic activities, which are more intensive on workdays than on holidays. It is worth mentioning
that the overall contribution of the NPF to particle number concentrations is less than what is seemingly
indicated by the diurnal patterns alone since NPF events occur on approximately 20% of days (Table 2).



Emissions from vehicular road traffic is represented by a notable peak during the morning rush hours
(between 05:30 and 08:30) on workdays. It is noted that the boundary layer mixing height is usually
increased during this interval because of the increasing solar radiation intensity and mixing intensity.
Another peak occurred around 21:00, thus later than the afternoon rush, which usually happens between
16:30 and 18:30. Under strong anti-cyclonic conditions, the evolution of the boundary layer mixing
height and mixing intensity can decrease the concentration levels in the afternoons until sunset, and this
can compensate the increased intensity of emissions. This all means that the afternoon peak is realised
in a fuzzy manner since it is more influenced by local meteorology than by vehicular emissions. The
effect of residential heating and combustion activities at evenings can also play a role. It is worth noting
that the early-morning rush-hour peak on event days was smaller than on non-event days, which agrees
with our earlier observation derived directly from experimental data (Salma et al., 2017) and is in line
with the overall picture on urban NPF events (Zhang et al., 2015; Kulmala et al., 2017). On holidays,
the modelled diurnal variation for non-event days contained an increasing part in the morning to a
modest concentration level, which remains fairly constant over the daytime. This is explained by the
differences in daily activities of citizens on workdays and holidays as far as both their intensity and
timing are concerned.

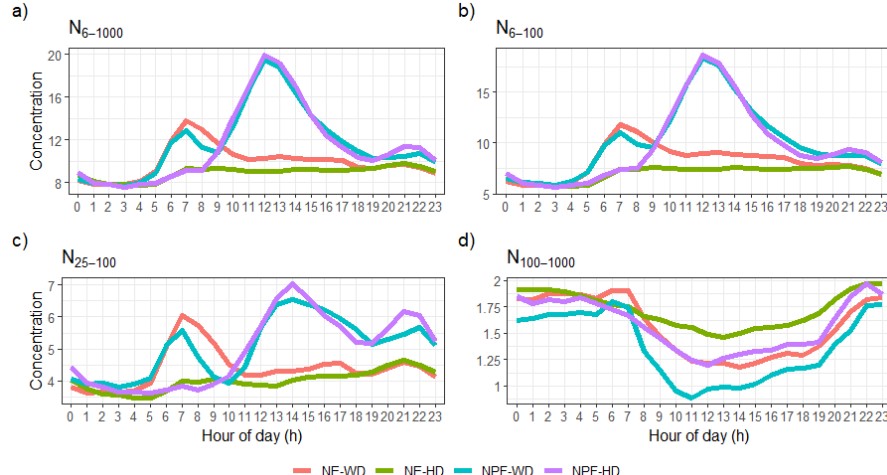


**Figure 3.** Diurnal patterns of particle number concentrations in the diameter ranges from 6 to 1000 nm ($N_{6-1000}$),
from 6 to 100 nm ($N_{6-100}$), from 25 to 100 nm ($N_{25-100}$) and from 100 to 1000 nm ($N_{100-1000}$) in units of $10^3$ cm$^{-3}$.
Red: non-event on workdays, green: non-event on holidays, cyan: event on workdays, purple: event on holidays.
The statistical diurnal patterns of $N_{6-1000}$ trends (Fig. 3a) were very similar or analogous to those of the
$N_{6-100}$. These two size fractions are strongly connected with each other as explained in Sect. 3.1. The
diurnal curves for $N_{25-100}$ (Fig. 3c) were also similar to the previous corresponding curves as far as the
character and shape are concerned, while there were also evident differences between their relative





structures. The peaks for the early morning and late afternoon rush hours were relatively larger than in
the trends of 6–100- or 6–1000-nm size fractions due to the higher contribution of primary particles
from high temperature sources in this size fraction. New particle formation generally occurs on days
when $N_{25–100}$ are smaller before the event onset (between 08:00 and 11:00). The maximum of the peaks
associated with NPF events in Fig. 3a and b – which is between 12:00 and 13:00 – was also shifted to
later, i.e. to ca. 14:00 in Fig. 3c. This can be explained by the time needed for freshly nucleated particles
to reach the diameter range >25 nm.
The statistical diurnal patterns for $N_{100–1000}$ seem very different from the smaller size ranges. First, their
time variations were rather small in comparison to the other size fractions. On workdays, they only
showed a modest elevation from 06:00 to 08:00 (morning rush hours), which is mainly caused by
resuspension of road/surface dust particles by moving vehicles or by emissions of coarse particles from
material wear. This morning peak was even missing on holidays, but another small and broad elevation
showed up from 21:00 to 22:00. This and the overall changes during the daylight time are primarily
related to the daily cycling of local meteorological conditions, in particular of boundary layer mixing
height under stabile anti-cyclonic weather conditions, outlined above.
**3.2.2 Effects of variables**
Monthly mean coefficients (mean $v_m$ slopes in Eq. 4) of $NO_2$, $O_3$ and $SO_2$ derived by GLMM, which
express their partial effects on particle number concentrations are shown in Fig. 4 for different size
fractions.

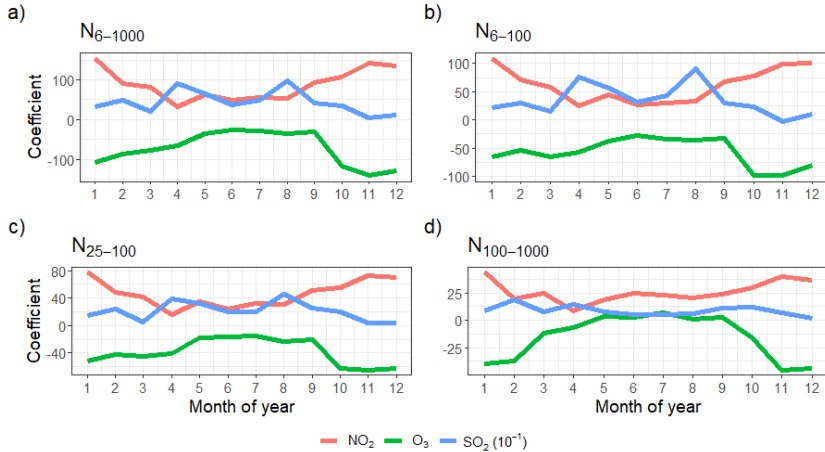


**Figure 4.** Distribution of monthly mean coefficients (which are proportional to the partial effects) for $NO_2$, $O_3$ and
$SO_2$ on particle number concentrations separately in the diameter ranges from 6 to 1000 nm ($N_{6–1000}$), from 6 to
100 nm ($N_{6–100}$), from 25 to 100 nm ($N_{25–100}$) and from 100 to 1000 nm ($N_{100–1000}$).



The coefficients of $SO_2$ and $NO_2$ are positive, while $O_3$ seems to have a decreasing effect on particle
number concentrations. The coefficients all have seasonal patterns, which means that the magnitude of
their effect on particle concentrations are of different magnitude in different months. This means for
example that 1 μg m$^{-3}$ increase in $NO_2$ concentration increases $N_{6-1000}$ concentration in January by 154
m$^{-3}$ but in June by 50 m$^{-3}$. This could, however, be partly caused by annual changes of boundary layer
mixing height or some other variable affecting particle concentrations, and correlating with these, but
not measured at the site. The boundary layer mixing height tends to be smaller in Budapest in winter
than in the other seasons (Salma et al., 2011), which ordinarily results in higher atmospheric
concentrations at steady-state absolute amounts of chemical species. The coefficients of $NO_2$ on $N_{6-1000}$,
$N_{6-100}$ and $N_{25-100}$ were higher in winter. This may indicate that large fractions of particles in these three
size fractions originate from residential heating and $NO_2$ acts as an indicator for this source. Another
major source of $NO_2$ and primary particles is the road traffic, but this does not show seasonal variation
in Budapest. The seasonal effect of $NO_2$ on chemically aged, regional type particles ($N_{100-1000}$) may not
be significant.
The partial effect of $O_3$ on $N_{6-1000}$, $N_{6-100}$ and $N_{25-100}$ was weaker in summer, late spring and early autumn.
This interval coincides with relatively large $O_3$ concentrations in the area. Ozone has a strong seasonal
variation (as shown in Fig. S1 in the Supplement). The negative correlation between $O_3$ concentration
and its effect on particle concentrations need further clarification since $O_3$ participates in a large variety
of complex atmospheric processes and also serves as a marker for photochemical processes which
influence secondary particle formation. The influence of $O_3$ on $N_{100-1000}$ was virtually negligible due
likely to the regional character of these particles (which are usually chemically aged and often represent
larger spatial scale due to their larger atmospheric residence time) similarly to $NO_2$. In addition, $O_3$
might act as an indicator of particulate pollution from traffic, power plants and other anthropogenic
sources. Then more ozone would indicate higher number of larger particles and due to coagulation less
smaller particles.
The partial effects of $SO_2$ on the particle number concentrations were the largest of the three gases
considered. In the $N_{6-1000}$, $N_{6-100}$ and $N_{25-100}$, two peaks appeared, one in spring and another one in late
summer. This shape is in line with the average distribution of the monthly mean relative NPF occurrence
frequency in Budapest (Salma and Németh, 2019). The latter distribution consists of an absolute and a
local minimum in January (with a monthly mean occurrence frequency of 5.9%) and in August (17.0%),
respectively, and an absolute and a local maximum in April (41%) and in September with (26%),
respectively. The distribution of the $SO_2$ coefficient suggests and confirms that $SO_2$, via NPF events
contribute in a substantial extent to the particle number concentrations in cities. The influence of $SO_2$
on $N_{100-1000}$ was virtually negligible due likely to the regional character of these particles similarly to the
other two gases included into the model.



Figure 5 summarizes the effect of macro-circulation patterns on particle number concentrations in the
different size fractions. It is seen that the larger regional-type particles are less affected by the MCPs
than the smaller particles. The weather conditions favouring NPF events can be identified from the
curves by looking at the largest coefficients for size fraction of 6–100 nm.





It seems that the MCP no. 3 (Mediterranean cyclone with a cold front over S Europe, N wind), 7 (highly
developed cyclone over N Europe, W wind) and 12 (anticyclone over the Carpathian Basin, changing
wind direction) can represent favourable conditions for NPF events than the other MCPs. Under these
conditions, the weather in the area is typically windy, with average solar radiation (expect for MCP no.
3 in summer when it shows low daily values), with strong planetary bounding layer evolution and
consequently, iv) the pollutants concentrations are below the average (expect for the winter inversions
in MCP no. 12). The air pollution situations are better separated by MCP codes in summer than in winter.
The weather type classified as no. 6 (Mediterranean cyclone with a warm front over S Europe, S wind)
disfavour the events. Under these conditions, the weather is typically cloudy and rainy with lower than
average solar radiation. This situation is often associated with polluted air in Budapest. Proportions for
NPF days for different MCP codes, which are shown Table S1 in the Supplement, also confirm these
conclusions.

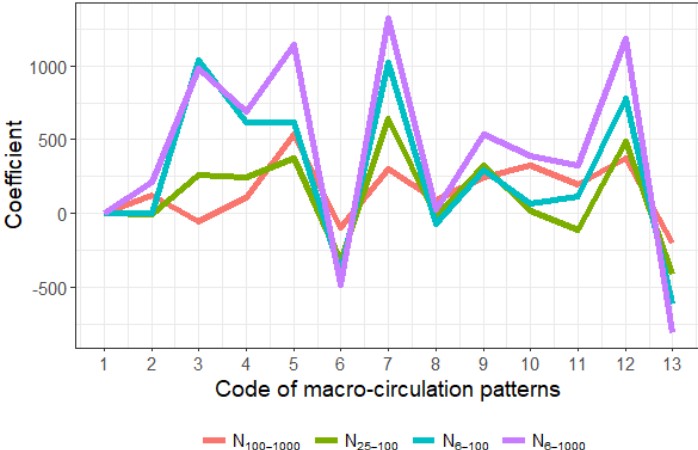

**Figure 5.** Distribution of monthly mean coefficients (which are proportional to the partial effects) for macro-
circulation patterns (Péczely codes) on particle number concentrations separately in the diameter ranges from 100
to 1000 nm ($N_{100-1000}$), from 25 to 100 nm ($N_{25-100}$), from 6 to 100 nm ($N_{6-100}$) and from 6 to 1000 nm ($N_{6-1000}$).
The coefficients for GRad and RH for different size fractions are shown in Fig. 6. It was found that these
variables do not have seasonal dependency, i.e. they contribute with equal strength to particle
concentrations throughout the year. Effect of GRad is positive for all size fractions, but it is weaker for
larger (regional-type or already chemically aged or processed) particles. The latter contribution could
be related to the bias in meteorological properties as well. The RH has negligible effect on size fraction
of 25–100 nm. It affects strongly and positively the largest particles, which means that the particles are
larger within higher humidity. This might be related to local meteorology, as higher RH probably means
more clouds and more clouds probably means less radiation and lower boundary layer and this could





cause higher particle concentration. In contrast, the effect of RH on the smallest particles was negative,
which is probably caused by high RHs, which limit NPF (e.g. Hamed et al., 2011).

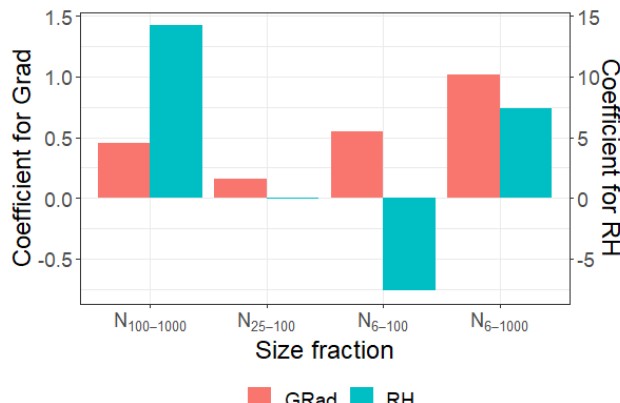

**Figure 6.** Distribution of monthly mean coefficients (which are proportional to the partial effects) for global
radiation (GRad) and relative humidity (RH) separately in the diameter ranges from 100 to 1000 nm ($N_{100-1000}$),
from 25 to 100 nm ($N_{25-100}$), from 6 to 100 nm ($N_{6-100}$) and from 6 to 1000 nm ($N_{6-1000}$).

### 3.2.3 Goodness-of-fit evaluation for GLMM

In order to estimate the uncertainty of the models for different size fractions, we calculated the mean
absolute errors relative to the dependent variable mean, given by Willmot et al. (2009):
$$Err = (n^{-1} \sum_{i=1}^{n} |y_i - \hat{y}_i|) \cdot \bar{y}^{-1} \,, \tag{5}$$
where $n$ is the number of observations, $y_i$ are the observed particle number concentrations, $\hat{y}_i$ are the
predicted values given by the GLMM and $\bar{y}$ is the mean of the observed values. In addition, we
calculated Spearman's rank correlation coefficients between the observed and predicted values for all
size fractions. Both goodness-of-fit estimates are shown in Table 4. As the relative errors for different
size fractions are within a range of 0.30–0.34 and the correlations are higher than 0.70, it can be
concluded that the model fitted the data with this size and measurement uncertainty well.

**Table 4.** Goodness-of-fit estimates for GLMM as expressed by the mean absolute error relative to the dependent
variable mean and by Spearman's rank correlation coefficient separately in the size fractions of 6–1000, 6–100,
25–100 and 100–1000 nm.

| Size fraction | Error | Correlation |
|---|---|---|
| 6–1000 | 0.30 | 0.73 |
| 6–100 | 0.32 | 0.72 |
| 25–100 | 0.34 | 0.71 |
| 100–1000 | 0.34 | 0.73 |


Figure 7 illustrates how well the GLMM model predicts the observations in all size fractions within a
randomly selected period of one week in March 2015. The figure shows that the predicted values follow
the observations fairly well in all size fractions. Overall, the statistical model finds the peaks of the
concentration, but slightly underestimates the highest peaks and the fastest fluctuations and in some
cases, overestimates the lowest concentrations.

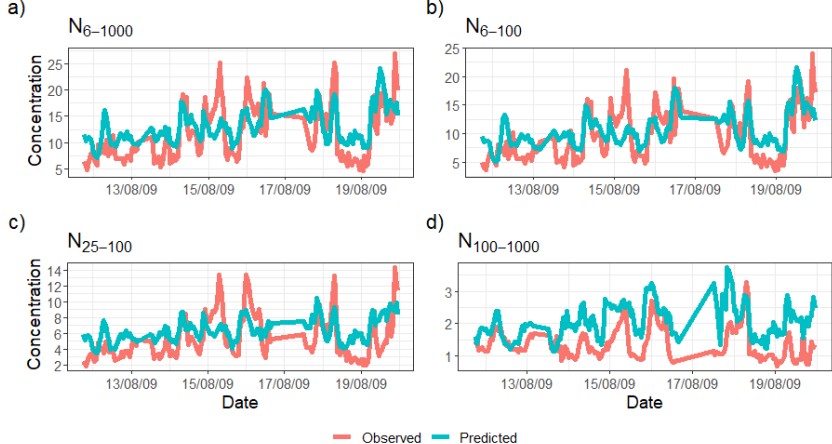


**Figure 7.** Observed (red line) and predicted (cyan line) time series for an illustrative example period separately in
the size fractions of 6–1000, 6–100, 25–100 and 100–1000 nm.
**4 Conclusions**
In the present study, we determined decennial statistical time trends and diurnal statistical patterns of
atmospheric particle number concentrations in various relevant size fractions in the city centre of
Budapest in an interval of 2008–2018. The decennial statistical trends showed decreasing character
except for the size fraction representing the regional. The mean overall decrease rate was approximately
–5% scaled for the 10-year measurement interval. The decline can be interpreted as a consequence of
the decreased anthropogenic emissions in the city. The diurnal statistical patterns suggested that reduced
traffic emissions were most likely an important factor in causing the observed changes. It is expected
that traffic intensity changed in a modest manner in the city centre during the time interval of interest,
so our results indicate that the reductions is most likely related to lower emission factors. This appears
to follow some changes of sulfur content in fuels and control measures on emissions for on-road heavy-
duty diesel vehicles. Introduction of better particle filters in diesel cars, cleaner fuel and more
sophisticated diesel engines could also contribute. The changes appear to have responded to both the
policy on urban air quality and the influence of economic circumstances of inhabitants. Excitingly, the
mean ages of passenger cars and busses in Hungary increased during the years under investigation. The
exact explanation of the decrease requires continuation of the related measurements and further


dedicated studies. The present results can be also used for evaluating the effectiveness of present and
prospective mitigation policies.
The diurnal statistical patterns can be also utilized in interpreting some properties of NPF events in urban
environments, and in explaining time evolution of particle number concentration. As a result of GLMM,
we could, for instance, give a parametrization for predicting particle concentrations in different size
fractions. Models similar to those developed in the present study could be used for other particle sizes or
locations as well. The same parameterization cold be used at least in areas with similar concentration
levels of particles and pollutants, while the extrapolation of the results to cleaner or more polluted
environments needs to be confirmed before the use. Conjugate or linked parameterizations to be
developed for varying environments can be implemented as a part of atmospheric models to predict the
concentrations of climatically active particles in order to reduce their extensive computational times. In
addition, this could also contribute to solving some current uncertain issues in the theoretical description
of NPF and growth process, particularly when predicting cloud condensation nuclei concentrations.
**Data availability.** The observational data used in this paper are available on request from the corresponding author
Imre Salma.
**Author contributions.** IS and SM formulated the original concept; ZN, VV, TW and IS collected and processed
the experimental data; SM, VL and TY were responsible for the statistical data analyses and their physical basis;
SM and IS interpreted the results; IS and SM wrote the manuscript with contributions from all co-authors.
**Competing interest.** The authors declare that they have no conflict of interest.
**Financial support.** The research was supported by the National Research, Development and Innovation Office,
Hungary (contracts K116788, PD124283 and K132254), by the János Bolyai Research Scholarship of the
Hungarian Academy of Sciences (ZN) and by the European Regional Development Fund and the Hungarian
Government (GINOP-2.3.2-15-2016-00028), The Nessling Foundation, The Academy of Finland Centre of
Excellence (grant no. 307331) and The Academy of Finland Competitive funding to strengthen university research
profiles (PROFI) for the University of Eastern Finland (grant no. 325022) and Academy of Finland project (grant
no. 299544)

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
