# Peer review of "Decennial time trends and diurnal patterns of particle number concentrations in a Central European city between 2008 and 2018"

_Atmospheric Chemistry and Physics, 2020_

## Referee Comment (RC1) · Anonymous Referee #1 · 7 Jul 2020

I found the manuscript well written and scientifically sound. I only found one error/inconsistency in Figure 7: In the text it says that Figure 7 illustrates the GLMM model predicts the observations in all size fractions within a 543 randomly selected period of one week in March 2015, but in reality the time period is from 2013-2019.
* * *

---

## Referee Comment (RC2) · Anonymous Referee #2 · 7 Aug 2020

General comments

The paper presents convincing arguments with statistical methods as why it is likely that decreased traffic emissions have led to decreased particle pollution in Budapeszt. Hence, these results are extremely important from a policy making point of view. In other words, it is neccessary that there are no doubts as to why concentrations have gone down. For this reason, the paper must undergo additional analysis to prove the point that car emissions have indeed led to the decrease of particle concentrations in Budapeszt. If this analysis can not conclusively prove the reason for the decrease, then I must stress the need to be more careful about the conclusion in the abstract

and conclusion section as to the reason for the decrease. And provide a sentence that other reasons for the decrease can not be ruled out.

Hence, I suggest to accept the paper subject to major revision of data analysis as suggested below. In addition there are a few minor remarks that need to be addressed.

Major remarks

Table 2 seems to indicate that there are increasing number of cars and car age as function of time in Budapeszt. How this can lead to decrease in emissions from traffic in chapter 3.1 is not quite convincing despite that Euro regulations are imposing restrictions to emissions from new cars. The total emissions are likely dominated by the ageing cars, and the average age of cars is increasing with time. Is there still a likelihood that the statistical analysis could be wrong for some unknown reason, and the reason for decreasing concentrations is arising as function of varying meteorological conditions between different years as expressed by the MCP parameter and not due to decreased car emissions? After all, the gas concentrations are not decreasing with time as the authors admit. I want the authors to quantify how often MCP weather patterns of type 3, 7, and 12 occur during April, May and August during the earlier years compared to the later years, and see if this can explain why the N6-1000 is higher in the earlier years than the later years (and also for the MCP that do not favor NPF, type 6).

The statistical method of DLM and GLMM are not enough for interpreting the results. A manual analysis of the MCP as described above must be done as well to provide further evidence for decreasing particle trend as being caused by decreasing traffic emissions. I stress also to use an educated guess/calculation of how you expect to see reductions in emissions of particles based on the values of car age and number of vehicles and diesel car share as function of year, and see if your expecations indeed would indicate a decrease in emissions. Just because there is a tendency for decreasing car emissions in Germany, doesn't prove that the same thing is happening

for Budapeszt if the traffic fleet composition is different in Budapeszt. Also, I want the authors to look more carefully on meteorological parameters manually as well. Even though for example the average temperature is the same for the different years, doesnt' mean that it is not varying between different days that lead to the particle concentration trends between different years.

Minor remarks

Please denote that the station is an "urban background" station in the abstract, introduction and methodology sections. This is needed for other readers to relate to the expected pollution level, and to know if this is the most polluted place in Budapest, or as you have in this case, a medium population exposure location in the city center, so called urban background. Alternatively, if it is not a typical urban background site, but slightly more polluted (I don't know this), please explain in the introduction and methodology section that the site has a pollution level between a typical heavy trafficked street level site and an urban background site, but closer to typical urban background levels.

Lines 133-134. "the most extensive inter-comparison was realised in summer 2015 (Salma et al., 2016a) and autumn 2019". Please denote which kind of intercomparisons were made.

Chapter 2.1: SO2, CO, NO, NOx, O3, and PM10 measurements: What is the pollution level of the site measuring these paramters: urban background or street level pollution, or something inbetween, or cleaner than an urban background site? Please describe this station as well. Otherwise, we cannot compare the time trends for this site as compared to the BpArt site. For example, the BpArt site might be closer to traffic than the gas measurement site explaining why the BpArt concentrations are decreasing with decreasing traffic emission trends, but not at the gaseous concentration site, which is then relatively more influenced by background long range sources.

Chapter 2.2: The MCP codes are developed for 00:00 UTC time. When you have a time of your particle or gas concentration data or meteorological data, which is the

MPC type that you use? For example, if the measurement time is August 1, 14:00 local time, what is the MCP coding for that time? Is it the MCP coding from August 1, 00:00 UTC, or MCP coding from August 2, 00:00? Or is it denoted as a combination of both MCP codes? It should be clearly stated in the manuscript. The MCP coding from one day to the other might change completely, meaning MCP codes for a measurement time inbetween two MCP coding times can be ambiguous.

Chapter 2.3.1. Would you please explain the autoregressive component?

Equation (4): The MCP is not a continuous variable, but it is discrete. How can you construct a linear output factor from Beta-6 multiplied with MCP-i? Would you mind explaining how Beta-6 and MCP-i and their product are constructed?

Line 274: Q (GRad) calculation is incorrect. The equation is correct only if you have 100 % data coverage. You have between 90 and 100 % data coverage as indicated in the method section. Hence, the calculated value will be systematically underestimated unless you interpolate data for the missing hours of GRad data. This could potentially be the reason why Table 2 Q values are different for different years, and not due to varying total insolation during one year to the other.

Lines 317-318. It is a strong statement to say that "this decoupling confirms that the causes of the decrease in particle number concentrations are not primarily related to meteorological conditions because they would jointly affect the gas concentrations as well". That gas concentrations don't go down and particle number concentrations go down could by accident also be related to a difference in availability of different MCP days during different years and seasons. NPF events could be favored in earlier years due to for example quite high number of certain MCP days with lower particle surface area, which favours NPF, which don't appear as frequent in the later years. This could happen even if the median particle surface area is decreasing every year (as indicated by decreasing N100-N1000). But, this difference in MCP does not automatically mean that the gas concentrations should change in the same way as N6-1000. Hence, I

would rephrase the wording from "confirms" to "suggests".

Conclusion: You mention that the accumulation mode particles don't show a decreasing annual trend. But, according to Table 3 they do.

---

## Author Comment (AC1) · 25 Aug 2020

I found the manuscript well written and scientifically sound. I only found one error/inconsistency in Figure 7: In the text it says that Figure 7 illustrates the GLMM

model predicts the observations in all size fractions within a randomly selected period of one week in March 2015, but in reality the time period is from 2013-2019

Answer:

We thank the reviewer on the positive comment. Thank you for noticing that the Figure 7 was incorrect, the correct figure is now changed to the manuscript.

---

## Author Comment (AC2) · 25 Aug 2020

We thank the reviewer for the review and constructive comments. Below we address each comment point by point. Reviewer comments are marked as black, our response as blue and corrections to the changes to the manuscript as red.

**General comments**

The paper presents convincing arguments with statistical methods as why it is likely that decreased traffic emissions have led to decreased particle pollution in Budapeszt. Hence, these results are extremely important from a policy making point of view. In other words, it is neccessary that there are no doubts as to why concentrations have gone down. For this reason, the paper must undergo additional analysis to prove the point that car emissions have indeed led to the decrease of particle concentrations in Budapeszt. If this analysis can not conclusively prove the reason for the decrease, then I must stress the need to be more careful about the conclusion in the abstract and conclusion section as to the reason for the decrease. And provide a sentence that other reasons for the decrease can not be ruled out.

A: As the reviewer pointed out, we cannot be certain that the decreasing particle numbers are due to traffic emissions. Improvements in heavy-duty vehicle fleet has been done, as stated in the conclusion and that certainly affects. Total number of passenger cars has increased, though the fraction of diesel cars from the total number of passenger cars has been increasing. Diesel engines have smaller particle emissions than gasoline engines as shown e.g. in Wihersaari et al. (2020) which can be assumed to lead smaller total particulate emissions of passenger car fleet. The sentence in the abstract is reformulated as:

This was interpreted as a consequence of the decreased anthropogenic emissions at least partly from road traffic alongside to household heating and industry.

Hence, I suggest to accept the paper subject to major revision of data analysis as suggested below. In addition there are a few minor remarks that need to be addressed.

**Major remarks**

Table 2 seems to indicate that there are increasing number of cars and car age as function of time in Budapeszt. How this can lead to decrease in emissions from traffic in chapter 3.1 is not quite convincing despite that Euro regulations are imposing restrictions to emissions from new cars. The total emissions are likely dominated by the ageing cars, and the average age of cars is increasing with time. Is there still a likelihood that the statistical analysis could be wrong for some unknown reason, and the reason for decreasing concentrations is arising as function of varying meteorological conditions between different years as expressed by the MCP parameter and not due to decreased car emissions? After all, the gas concentrations are not decreasing with time as the authors admit. I want the authors to quantify how often MCP weather patterns of type 3, 7, and 12 occur during April, May and August during the earlier years compared to the later years, and see if this can explain why the N6-1000 is higher in the earlier years than the later years (and also for the MCP that do not favor NPF, type6).

A: Our present interpretation of the reduced traffic emissions cannot be completely conclusive in all aspects and further investigations and collection of new data are planned on the basis of the present results. As the GLMM model accounts for the meteorological patterns on daily level, it is highly unlikely that the changes in occurrence of certain MCP patterns could cause the decreasing trend in particle numbers. To ensure this, we studied the occurrence of the MCP patterns in detail and we saw no significant change in frequencies of any of the patterns, specifically in the NPF favouring or disfavouring ones pointed out by the reviewer. In addition, MCP types 3 and 7 are rather rare (Table 1), and thus cannot have significant effect on trends in total. The following sentences are added to the manuscript section 3.2.2

In order to see if the decreasing concentrations are due to changes in meteorological patterns, we investigated separately the occurrence of the MCP patterns during the measurement period. We found no significant changes in the occurrence of the patterns and thus the decreasing particle concentrations are due to something else than the meteorological patterns.

The statistical method of DLM and GLMM are not enough for interpreting the results. A manual analysis of the MCP as described above must be done as well to provide further evidence for decreasing particle trend as being caused by decreasing traffic emissions. I stress also to use an educated guess/calculation of how you expect to see reductions in emissions of particles based on the values of car age and number of vehicles and diesel car share as function of year, and see if your expecations in-deed would indicate a decrease in emissions. Just because there is a tendency for decreasing car emissions in Germany, doesn't prove that the same thing is happening for Budapeszt if the traffic fleet composition is different in Budapeszt. Also, I want the authors to look more carefully on meteorological parameters manually as well. Even though for example the average temperature is the same for the different years, doesnt' mean that it is not varying between different days that lead to the particle concentration trends between different years.

A: We do not state that the decrease is only from traffic emissions, and with the text modifications above we try to clarify that. However, even though there might be certain differences between the composition of passenger car fleet between Germany and Budapest, the traffic emissions can still be assumed to be decreased also in Hungary even though the renewal rate of the fleet might be slower. We do not have exact details on the car fleet composition, and we do not have the expertise to make such educated guesses suggested by the reviewer. Thus, we can only speculate the magnitude of the effect of traffic emission reduction but, as stated above, the direction is clear.

**Minor remarks**

Please denote that the station is an "urban background" station in the abstract, introduction and methodology sections. This is needed for other readers to relate to the expected pollution level, and to know if this is the most polluted place in Budapest, or as you have in this case, a medium population exposure location in the city center, so called urban background. Alternatively, if it is not a typical urban background site, but slightly more polluted (I don't know this), please explain in the introduction and methodology section that the site has a

pollution level between a typical heavy trafficked street level site and an urban background site, but closer to typical urban background levels.

A: The requested information was added to the Abstract (1) (lines 11-13) and this point was also further emphasized in the body text (lines 115-117) where the measurement site is described (2).

(1) Multiple atmospheric properties were measured semi-continuously in the Budapest platform for Aerosol Research and Training Laboratory, which represents the urban background for a time interval of 2008-2018.

(2) This location represents a well-mixed, average atmospheric environment for the city centre of Budapest due to its geographical and meteorological conditions (Salma et al., 2016a), thus it can be regarded as an urban background site.

Lines 133-134. "the most extensive inter-comparison was realised in summer 2015(Salma et al., 2016a) and autumn 2019". Please denote which kind of intercomparisons were made.

A: The explanation was extended as requested. The following text was inserted to Page 4 lines 136-142.

First, the measured data by the CPC deployed in the DMPS system were compared to that of an identical stand-alone CPC operated in parallel. The agreement between the instruments was in accordance with the nominal specification of CPCs. In the next step, the integrated concentrations obtained from the size-resolved DMPS data were compared to the concentrations measured directly by the stand-alone CPC. The two instruments were again operated in parallel. The median CPC/DMPS ratio was utilised as correction factor for particle diffusion losses in the DMPS system (Salma et al., 2016a).

Chapter 2.1: SO2, CO, NO, NOx, O3, and PM10 measurements: What is the pollution level of the site measuring these paramters: urban background or street level pollution, or something in between, or cleaner than an urban background site? Please describe this station as well. Otherwise, we cannot compare the time trends for this site as compared to the BpArt site. For example, the BpArt site might be closer to traffic than the gas measurement site explaining why the BpArt concentrations are decreasing with decreasing traffic emission trends, but not at the gaseous concentration site, which is then relatively more influenced by background long range sources.

A: Further details on the character of the monitoring station for criteria pollutants was added as requested on page 5, lines 157-160.

This station ordinarily measures the smallest levels of the criteria air pollutants among the four monitoring stations located in the city centre. It can, therefore, be considered to represent the air pollution in between the urban background and street level/kerbside site.

Chapter 2.2: The MCP codes are developed for 00:00 UTC time. When you have a time of your particle or gas concentration data or meteorological data, which is the MPC type that you use? For example, if the measurement time is August 1, 14:00 local time, what is the MCP coding for that time? Is it the MCP coding from August 1, 00:00UTC, or MCP coding from August 2, 00:00? Or is it denoted as a combination of both MCP codes? It should be clearly stated in the manuscript. The MCP coding from one day to the other might change completely, meaning MCP codes for a measurement time in between two MCP coding times can be ambiguous.

A: The MCP codes represent the macro-circulation conditions in the whole Carpathian Basin as a geographical unit, and are determined on a daily bases in a discrete manner. They were assigned to the concentration and meteorological data of the whole given day. This is now clarified in page 6 line 191 in the revised manuscript:

Thus defined MCP was assigned to the following day in the data.

Chapter 2.3.1. Would you please explain the autoregressive component?

A: Autoregressive component is a parameter taking account the autocorrelation in the data which means that subsequent (here daily) measurements of variables are correlated. We added the following description to section 2.3.1, page 8, lines 232-234 in the revised manuscript (1) and added a new equation 4, describing the autoregressive component with description (2) on lines 242-245:

(1) The autoregressive component is added to the model in order to take account the autocorrelation in the data, i.e. the correlation between subsequent observations. Here it refers to first order autoregressive model (AR(1)).

(2) $\eta_t = \rho\eta_{t-1} + \varepsilon_{AR}, \varepsilon_{AR} \sim N(0, \sigma_{AR}^2)$,    (4)

where $y_t$ is the investigated concentration at time $t$, $\mu_t$ is the mean level and $\alpha_t$ is the change in the level from time $t-1$ to time $t$, $\gamma_t$ is the seasonal component, $\eta_t$ is an autoregressive error component and $\rho$ is the coefficient for autoregressive component, here fixed to $\rho = 0.6$.

Equation (4): The MCP is not a continuous variable, but it is discrete. How can you construct a linear output factor from Beta-6 multiplied with MCP-i? Would you mind explaining how Beta-6 and MCP-i and their product are constructed?

A: All levels of the MCP coding will get their own characteristic level for typical conditions calculated with the GLMM and thus $\beta_6$ is a (13×1) vector of coefficients. The levels of these coefficients are shown in Figure 5. To clarify this, we modified the description of $\beta_6$ in Eq. 5 (note changed numbering due to changes above) to following form (page 9, lines 282-284):

…, $\beta_6$ is the (*13*×1) vector of coefficients for different macro-circular patterns (MCP) indicating the characteristic level of number concentration during each MCP type, which are treated here as categorical variable…

Line 274: Q (GRad) calculation is incorrect. The equation is correct only if you have 100 % data coverage. You have between 90 and 100 % data coverage as indicated in the method section. Hence, the calculated value will be systematically underestimated unless you interpolate data for the missing hours of GRad data. This could potentially be the reason why Table 2 Q values are different for different years, and not due to varying total insolation during one year to the other.

A: The calculation method of the annual insolation was revised and improved taking into account the reviewer comment. As there were no big gaps in radiation measurements the missing data were interpolated and the calculation schema was also modified. The corresponding text (page 10, lines 289-293) and values in Table 2 were amended accordingly.

Annual insolation ($Q$), which expresses the total energy density at the receptor site, was derived from the individual hourly mean $GRad_{i,i}$ data, where index $i$ represents the hour of day (from 0 to 23), index $j$ stands for the day of year (from 1 to 365) as $Q = 3.6 \times 10^{-6} \times \sum_{i,j} GRad_{i,j}$. The dimensions of the $GRad_{i,j}$ data and $Q$ are W m$^{-2}$ and GJ m$^{-2}$ y$^{-1}$, respectively. The few randomly missing datapoints were interpolated linearly.

Lines 317-318. It is a strong statement to say that "this decoupling confirms that the causes of the decrease in particle number concentrations are not primarily related to meteorological conditions because they would jointly affect the gas concentrations as well". That gas concentrations don't go down and particle number concentrations go down could by accident also be related to a difference in availability of different MCP days during different years and seasons. NPF events could be favored in earlier years due to for example quite high number of certain MCP days with lower particle surface area, which favours NPF, which don't appear as frequent in the later years. This could happen even if the median particle surface area is decreasing every year (as indicated by decreasing N100-N1000). But, this difference in MCP does not automatically mean that the gas concentrations should change in the same way as N6-1000. Hence, I would rephrase the wording from "confirms" to "suggests".

A: Wording rephrased as suggested.

Conclusion: You mention that the accumulation mode particles don't show a decreasing annual trend. But, according to Table 3 they do.

A: The sentence is rephrased as (page 21, lines 579-580):

The decennial statistical trends showed decreasing character in all applied size fractions of particle concentrations.

New references

Wihersaari H., et al. (2020) Particulate emissions of a modern diesel passenger car under laboratory and real-world transient driving conditions, Environmental Pollution, 265, 114948, https://doi.org/10.1016/j.envpol.2020.114948.

Platt, S. M., El Haddad, I., Pieber, S. M., Zardini, A. A., Suarez-Bertoa, R., Clairotte, M., Daellenbach, K. R., Huang, R.-J., Slowik, J. G., Hellebust, S., Temime-Roussel, B., Marchand, N., de Gouw, J., Jimenez, J. L., Hayes, P. L., Robinson, A. L., Baltensperger, U., Astorga, C., and Prévôt, A. S. H.: Gasoline cars produce more carbonaceous particulate matter than modern filter-equipped diesel cars, Sci. Rep., 7, 4926, 2017, https://doi.org/10.1038/s41598-017-03714-9.